# Identification of Predictive Biomarkers of Lameness in Transition Dairy Cows

**DOI:** 10.3390/ani14142030

**Published:** 2024-07-10

**Authors:** Ana S. Cardoso, Alison Whitby, Martin J. Green, Dong-Hyun Kim, Laura V. Randall

**Affiliations:** 1School of Veterinary Medicine and Science, University of Nottingham, Sutton Bonington Campus, Sutton Bonington, Leicestershire LE12 5RD, UK; 2Centre for Analytical Bioscience, Advanced Materials & Healthcare Technologies Division, School of Pharmacy, University of Nottingham, Nottingham NG7 2RD, UKdong-hyun.kim@nottingham.ac.uk (D.-H.K.)

**Keywords:** liquid chromatography–tandem mass spectrometry, lameness, dairy cow

## Abstract

**Simple Summary:**

Lameness, an impaired gait, results from the pain dairy cows feel, which impacts their behaviour and performance, and consequently farmer income. Mobility scoring by visual inspection is the current gold standard in the UK for detecting this condition; however, it is prone to scoring biases and can only detect lameness once the cow feels pain, which may come too late for effective action to be taken. Conversely, the early detection of lameness allows for the better management and treatment of lameness cases, which is decisive in minimizing economic losses and animal pain, thereby improving cow welfare and the sustainability of the dairy industry. In a previous study, metabolites predictive of lameness were identified; however, the annotation of these metabolites has not yet been conducted. Although some annotation ambiguity is acceptable when performing untargeted analysis, accurately identifying compounds is crucial to understand which pathways are differentially altered between lame and non-lame cows. This study aimed to annotate the metabolites identified as predictors of lameness in the transition period and understand their biological role in the pathways and processes that lead to lameness. As a result, three metabolites predictive of lameness in dairy cows were identified at the highest confidence level, thus allowing for the understanding of the interactions between metabolism and immunity in lameness to be improved.

**Abstract:**

The aim of this study was to identify with a high level of confidence metabolites previously identified as predictors of lameness and understand their biological relevance by carrying out pathway analyses. For the dairy cattle sector, lameness is a major challenge with a large impact on animal welfare and farm economics. Understanding metabolic alterations during the transition period associated with lameness before the appearance of clinical signs may allow its early detection and risk prevention. The annotation with high confidence of metabolite predictors of lameness and the understanding of interactions between metabolism and immunity are crucial for a better understanding of this condition. Using liquid chromatography–tandem mass spectrometry (LC-MS/MS) with authentic standards to increase confidence in the putative annotations of metabolites previously determined as predictive for lameness in transition dairy cows, it was possible to identify cresol, valproic acid, and gluconolactone as L1, L2, and L1, respectively which are the highest levels of confidence in identification. The metabolite set enrichment analysis of biological pathways in which predictors of lameness are involved identified six significant pathways (*p* < 0.05). In comparison, over-representation analysis and topology analysis identified two significant pathways (*p* < 0.05). Overall, our LC-MS/MS analysis proved to be adequate to confidently identify metabolites in urine samples previously found to be predictive of lameness, and understand their potential biological relevance, despite the challenges of metabolite identification and pathway analysis when performing untargeted metabolomics. This approach shows potential as a reliable method to identify biomarkers that can be used in the future to predict the risk of lameness before calving. Validation with a larger cohort is required to assess the generalization of these findings.

## 1. Introduction

Lameness is a complex, painful condition with multifactorial aetiology mostly stemming from infectious and non-infectious hoof lesions, both likely to alter the lame cow’s metabolome [1,2]. This condition affects the welfare of dairy cattle, commonly resulting from pathology in the foot [3,4], with a reported prevalence in Great Britain of approximately 30% [5,6]. Lameness has major implications for production efficiency and sustainability due to impacts on production [7,8], reproduction [9], and the early culling of cows [8,10]. Together with increased veterinary and trimming costs [11], lameness negatively impacts farm economics [12]. Early detection is of utmost importance to reduce the negative impacts associated with lameness and improve treatment outcomes, thereby reducing its prevalence [8,13]. In 2021 [14], guidelines were outlined to implement early detection and prompt and the effective treatment (EDPET) of new cases of lameness to break the cycle of chronicity. However, this approach is dependent on the reliable and accurate identification of early lameness, which is challenging using current methods, and the use of predictive biomarkers may be a tool that in future can provide an earlier alert to lameness risk [1]. It is also well recognized that mobility scoring alone to detect digital dermatitis is insufficient. It is known that the transition period, the periparturient period between 3 weeks before and 3 weeks after calving, is a critical phase triggering metabolic alterations thought to be associated with subsequent lameness risk that can be measured in urine metabolic profiles [1,15,16]. These metabolic changes appear to be associated with the stress caused by calving and preparation for lactogenesis, inducing physiological changes (metabolic and endocrine) that affect immunity [15,17], although fundamental knowledge gaps still exist concerning the pathogenesis of lesions that cause lameness and the effect of the transition period on pathological pathways.

Metabolomics is a cutting-edge scientific field involving the measurement of low-molecular-weight molecules (metabolites) in biological specimens [18]. Its use has been increasingly adopted to diagnose complex metabolic diseases and identify disease biomarkers [19]. The urine metabolome, in particular, provides an important source of information about physiological states in response to disease manifestations [20]. However, the correlation of statistical variables to biochemical functions of metabolomics relies on accurate metabolite identification [21,22,23]. Metabolite identification is a fundamental tool to link the raw data generated by metabolomics studies into the biological context [24,25]. Before performing biochemical pathway analysis and formulating a hypothesis, it is crucial to increase confidence in the proposed metabolite identification method [26].

The use of urine (urinalysis) to assess the health status of dairy cows has already been proven to be a non-invasive diagnostic method that could be routinely used [27,28]. Previous studies have shown that metabolomics analysis allows for the detection of potential metabolites that discriminate the metabolic profiles of lame and healthy cows using urine [1,16], milk [29,30], and blood [15,17]. Randall et al. [1] used liquid chromatography–mass spectrometry (LC-MS) and machine learning (ML), applying a suite of five algorithms (support vector machine (SVM), elastic net regression, partial least squares (PLS) regression, random forest (RF), and multivariate adaptive regression splines (MARS)) to predict lameness. In this study, SVM was the best-performing ML predictive model with a mean accuracy of 82% at the time of lameness. In He et al.’s work [29], a set of four of the same algorithms, except for MARS, was also applied, achieving accuracies of 100% for RF (with 15 selected variables) and 95% for Elastic net, PLS, and SVM (with 10 selected variables). Using a targeted approach, Zhang et al. [15] found that increased serum concentrations of cytokines released into the circulation in cows that developed lameness postpartum, compared with the control group of cows, were evident at 4 and 8 weeks before calving. These findings suggest a subclinical phase to lameness, making the use of biomarkers to predict lameness an interesting approach. Potential biomarkers of lameness were also identified in a study on transition dairy cows using targeted quantitative metabolomics [16]. To assess the analysis of the biomarker performance of the top five metabolites, the authors used the area under the receiver operator characteristic curve (AUC), achieving accuracies between 0.98 and 1 (95% CI) for the different time points assessed. Although these are good indicators showing the predictive ability of these particular metabolites to potentially predict lameness, it is important to highlight that they were not the same across all time points, which makes it difficult to interpret the results. The transition period covers parturition, as well as the nutritional and environmental changes immediately after calving that are stress-inducing, causing metabolic changes [31]. The identification of metabolites that are predictive for lameness in the transition period enables the early prediction of cows at risk of lameness in that lactation, and the annotation and pathway analysis of these metabolites may provide valuable information relating to pathogenic pathways. The potential of untargeted LC-MS-based metabolite profiling and combined machine learning methods to predict lameness pre-calving and prior to clinical symptoms in first-lactation dairy cows with high accuracy has also been demonstrated, in addition to targeted methods [1]. The untargeted approach allows a more extensive analyte coverage when compared with targeted approaches [32]. Untargeted LC-MS analyses are open to hypothesis generation and biomarker discovery [33], and may find different pathways associated with lameness that have not been identified in the studies previously described, since targeted metabolite analysis focuses on a specific class of molecules or pathways.

Metabolite identification was not a focus of the study by Randall et al. [1], and therefore top-ranking mass ions were used in their predictive lameness model regardless of their ability to be identified and without knowing their biochemical roles in the cows. This is useful for a predictive model, but metabolite identification is crucial for understanding the underlying biological pathways associated with lameness. They noted that further work is needed to confirm the identity of mass ions identified in their study and that the pathway analysis of predictive metabolites would give important insights into the pathogenesis of lameness. This study aimed to identify the metabolites previously determined as predictors of lameness in the transition period [1] and to understand their biological role by conducting pathway analysis. Metabolic pathway analysis allows us to understand the interactions between metabolism and immunity in lameness.

## 2. Materials and Methods

### 2.1. Previous Generation of LC-MS Data

The data utilized in this study were generated from a previous case–control study [1], for which ethical approval was granted by the University of Nottingham Committee for Animal Research and Ethics (Reference No. 3120 200220 and 3132 200309). A detailed description of the methodology was previously reported [1].

A quality control (QC) sample was made by mixing equal volumes of all samples. This was used in the current work for accurate metabolite identification as a representative of 82 samples corresponding to Cohort 2 previously published by Randall et al. [1]. In summary, urine samples were collected from 67 Holstein dairy heifers (entering their first lactation) from one herd, who were subjected to weekly mobility scoring (0–3 scale, AHDB [34]) from 3 weeks pre-calving to 70 days in milk (DIM) to identify lame and non-lame cows. Samples were selected from lame and non-lame cows at three time points; pre-calving (PRE, *n* = 28), post-calving (POST, *n* = 28), and at the time of lameness (AT, *n* = 26). Cohort size was based on a pilot work published by He et al. [29] on dairy cows using a similar statistical analysis. Exclusion criteria were surgery or any treatment with antimicrobials or anti-inflammatories that the heifers had been subjected to in the current lactation. Urine was diluted and untargeted metabolomics data were obtained via LC-MS with MS/MS for the putative identification of the most abundant metabolites. A set of 5 ML models (SVM, elastic net regression, PLS regression, RF, and MARS) [1] was applied within the caret package [35] using R [36] to analyse the pre-processed datasets to create a list of important metabolites associated with lameness (Table 1). Recursive feature elimination was used for each of the 5 models to select the smallest number of variables (mass ions) that provided the best accuracy. Stability selection [37] was used to prevent overfitting and model triangulation was employed to reduce the probability of selecting false-positive variables, as previously described by Randall et al. [1].

### 2.2. Previous Metabolite Annotation and Identification

At the time of the previous LC-MS analytical run [1], metabolite identification was carried out by matching accurate masses and the polarity of the detected ion peaks with metabolites in the Bovine Metabolome Database [38], and retention times (RTs) with 268 authentic standards and/or ddMS/MS with the mzCloud fragmentation database (Thermo Fisher Scientific, Hemel Hempstead, UK) using Compound Discoverer 3.1 SP1 software (Thermo Fisher Scientific, Hemel Hempstead, UK). The confidence in the metabolite identification of the annotations resulting from pre-processing was then classified using a four-level scale (L1–4) based on the recommendation by the Chemical Analysis Working Group, Metabolomics Standards Initiative (MSI) [24,25]. In this classification, the metabolites identified using the mass-to-charge ratio (*m*/*z*), RT, and MS/MS with reference standards were classified as level 1 (L1); putatively annotated metabolites using *m*/*z*, RT, and/or MS/MS from the spectral library and no reference standards as level 2 (L2); putatively characterized metabolite classes as level 3 (L3); and unknowns as level 4 (L4).

### 2.3. Further MS/MS and Identity Confirmation with Standards

To further increase the confidence in the proposed annotations, LC-MS/MS with standards was performed. A QC sample was used for chemical analysis, performed in the same laboratory, using the same instrument (Dionex 3000 LC system coupled with Q-Exactive Plus mass spectrometer, Thermo Fisher Scientific, Hemel Hempstead, UK), and the same analytical methodologies as in previous LC-MS analyses [1].

A set of important mass ions selected from previous statistical analyses (Table 1) was used to create an inclusion list, which increased the number of metabolite identifications compared to the standard data-dependent acquisition [39]. Briefly, if a scanned ion *m*/*z* matched an ion in the inclusion list, within a 2 min RT range, it was then selected for MS/MS fragmentation [40]. The inclusion list was exported from the previous Compound Discoverer results dataset [1]. Data-dependent MS/MS was performed with a resolution of 17,500, maximum ion injection time (IT) of 50 ms, automatic gain control (AGC) target of 1 × 10^5^, isolation window of 0.5 *m*/*z*, and stepped normalized collision energies of 20, 30, and 40, both in ddMS/MS with others picked if idle, and in parallel reaction monitoring (PRM) mode with the inclusion list.

The first stage in the process of choosing the authentic standards to include in the LC-MS/MS analysis was a search for LC-MS spectra in the Bovine Metabolome Database (BMDB) [38] and the Livestock Metabolome Database [41] using the *m*/*z* (within 5 ppm tolerance), ion mode, and adduct type of the 37 important ions presented in Table 1. Then, the in-house availability of authentic standards in chemical libraries was searched, as illustrated in Figure 1. Authentic standards (Table 2) were used at a concentration of 20 µM.

Qual Browser in Xcalibur 2.2 SP1.48 (Thermo Fisher Scientific) was used to conduct an RT search of the *m*/*z* of each important predictive metabolite in all raw data files (QC and standards). Differences in retention times between QCs and standards were calculated and ions with differences less than or equal to 0.5 min were considered for further confirmation. Standard MS/MS spectra were input to mzVault (Thermo Fisher Scientific) to build a library, which was then used to compare the *m*/*z* and fragmentation spectra of the QC.RAW data files. Confidence in identification was checked using CFM-ID (Competitive Fragmentation Modelling for Metabolite Identification) [42], which uses in silico fragmentation techniques to predict ESI-MS/MS.

### 2.4. Enrichment and Pathway Analysis

Metabolite set enrichment analysis (MSEA) [43] was performed using the MetaboAnalyst 5.0 platform [44] to identify the biological pathways relating to the metabolites associated with lameness. Over-representation analysis (ORA) was performed using a list of 9 compounds of interest identified with L2 or L3 levels of confidence (Table 1), and the reference metabolome used as background was a list containing all the annotated compound names in the dataset (*n* = 524) [45]. The ORA used hypergeometric testing to assess whether compounds of interest were represented above what would be expected within the reference metabolome. RaMP (integrating HMDB/KEGG, Reactome, and WikiPathways) [46] was selected as the library pathway for this study and only metabolite sets containing at least three entries were considered. Both lists (9 compounds of interest and the reference metabolome) were previously subjected to compound name standardization in order to match the compound identity to the knowledge base used by MetaboAnalyst.

Another pathway analysis that integrated ORA and topology analysis was performed using the MetaboAnalyst 5.0 platform [44]. The pathway analysis parameters used were scatter plot (testing significant features) as the visualization method, the hypergeometric test for the ORA, and relative-betweenness centrality for the pathway topology analysis. All compounds in the Kyoto Encyclopedia of Genes and Genomes (KEGG) [47] *Bos taurus* (cow) pathway library were selected for the reference metabolome.

## 3. Results

A total of 37 mass ions found to be associated with lameness, based on previous work [1], were used for compound identification in this study. Before further MS/MS, one mass ion was classified (MSI classification of confidence in identification) as L2, eight as L3, and twenty-eight as L4. There were no ions classified as L1, the highest level of identification, as detailed in Table 1.

The comparison of all peaks’ *m*/*z* and RTs, obtained after further LC-MS/MS of the QC with the inclusion list and the standards library, resulted in 10 putative matches (Table 3). After comparison with standard MS/MS spectra, it was possible to increase the confidence of *m*/*z* 107.0500 (cresol) and *m*/*z* 177.0406 (D-glucono-delta-lactone) to level 1, and *m*/*z* 143.1080 (valproic acid) to L2 using mzVault. Sometimes there was an MS/MS match in mzVault, but because of the RT being different, the metabolite was not identified as this (e.g., gluconolactone). The predicted ESI-MS/MS using CFM-ID [42] also agreed with the measured MS/MS spectra for valproic acid in the QC sample.

The top 25 biological pathways impacted by the set of metabolites of interest resulting from the ORA are presented in Figure 2. When considering the threshold for a one-tailed *p*-value of <0.05, provided after adjusting for multiple testing [43], only six pathways were found to be significantly impacted, including branched-chain amino acid catabolism, glucose-6-phosphate dehydrogenase deficiency, ribose-5-phosphate isomerase deficiency, transaldolase deficiency, the valproic acid metabolism pathway, and the valproic acid pathway. Figure 3 shows the network view of all metabolite pathways created using the Enrichment Analysis module of MetaboAnalyst 5.0.

The metabolite of interest presented in the main enriched pathway, branched-chain amino acid catabolism, is ketoleucine (a standard for this metabolite was not available in-house). D-Gluconolactone is present in glucose-6-phosphate dehydrogenase deficiency, in ribose-5-phosphate isomerase deficiency, and in the transaldolase deficiency pathway. Only one match with the metabolites of interest was found for each enriched pathway.

The results from the pathway analysis combining ORA and topology analysis identified two pathways under the p-value threshold of 0.05: valine, leucine and isoleucine biosynthesis (*p*-value = 0.02), in which a single match of ketoleucine was present; and the pentose phosphate pathway (*p*-value = 0.04), in which D-gluconolactone was present.

## 4. Discussion

By increasing the confidence in metabolite identification and performing pathway analysis, this study aims to provide further insights into lameness pathogenesis, as its multifactorial aetiology and the involvement of animal-based and environmental risk factors have not been fully understood [48]. The method for the early detection and prevention of the disease in dairy cattle proposed by Randall et al. [1] showed promise to contribute to reducing lameness costs and increasing dairy cattle welfare, predicting lameness with an accuracy of 82% using ML models. The metabolites and the pathways now identified are the building blocks to developing early detection tools.

### 4.1. Metabolite Annotation and Identification

Overall, the obtained results allowed the identification of metabolites that had previously been determined as predictive for lameness in transition dairy cows. This study also highlighted the challenges of metabolite identification and pathway analysis with untargeted metabolomics due to limitations in published spectral libraries, which are discussed in Section 4.5.

### 4.2. Highest Level of Confidence: L1

After further MS/MS analysis, cresol (a synonym of 4-methylphenol) was confirmed at the highest confidence level of L1 (Table 3, Appendix A). In humans, free p-cresol is higher in patients hospitalized for infections [49] due to its ability to inhibit leukocytes from producing reactive compounds that play a role in eliminating infectious agents. In dairy cows, p-cresol was found to be increased in the urine of dairy cows with sub-clinical ketosis in a study conducted by Eom et al. [50], leading the authors to identify this metabolite as a potential biomarker for the diagnosis of ketosis. Significantly different levels of p-cresol were also observed in the cerumen samples between lame and control calves [51]. However, the authors make a reservation regarding the use of this potential biomarker and point out that more elucidation is needed, as it has already been found to be altered in cattle with other pathological conditions.

In our study, it was possible to identify cresol but not to distinguish which isomer was present, given that there are three isomers of cresol: ortho-cresol (o-cresol), meta-cresol (m-cresol), and para-cresol (p-cresol). To distinguish these isomers, nuclear magnetic resonance (NMR) spectroscopy would be required [52,53]. Thus, the structural identification of metabolites is carried out mainly by distinguishing MS/MS spectra, since the technique is theoretically capable of differentiating all metabolites. However, isomers generate very similar MS/MS spectra, making it difficult to distinguish between them [22]. Isomers, despite having the same molecular formula, possess a different structure and are widely distributed in organisms, often assuming completely different roles in metabolic pathways or even being present in different metabolic pathways [22]. The enrichment and pathway analyses performed in this study did not find any observed hits between this metabolite and the most enriched pathways. However, the results of the present study and the literature suggest that future research should include cresol to clarify its relationship with lameness.

Gluconolactone (*m*/*z* 177.0406), previously annotated as L3 (Table 1), increased in confidence of identification to L1 (Table 3, Appendix A). This metabolite was found to be associated with the pathways of glucose-6-phosphate dehydrogenase deficiency, ribose-5-phosphate isomerase deficiency, and transaldolase deficiency after ORA. The role of gluconolactone in the pathways in which it is involved is described in the section on enrichment and pathway analysis.

### 4.3. Next Level of Confidence: L2

Valproic acid increased in confidence of identification to L2 after matching the RT to the standard after further MS/MS analysis (Table 3), because the ion did not fragment in the collision cell after stepped normalized collision energies of 20, 30, and 40 V. Therefore, no fragmentation pattern match to any standard could be obtained. This metabolite was an observed hit in both the valproic acid metabolism pathway and the valproic acid pathway. A discussion about the role of valproic acid in these pathways is described below.

### 4.4. Enrichment and Pathway Analysis

The new LC-MS/MS analysis based on the previous study provided an increased confidence in the putative metabolite annotations predictive for lameness, enabling enrichment and pathway analysis to link changes in metabolic compounds to biological pathways in response to lameness. The pathway analysis performed in this experiment provided insights into the pathways that may be impacted by lameness; however, the results should be interpreted with care, since only one match with the metabolites of interest was found for each pathway. Branched-chain amino acid catabolism, glucose-6-phosphate dehydrogenase deficiency, ribose-5-phosphate isomerase deficiency, transaldolase deficiency, the valproic acid metabolism pathway, and the valproic acid pathway were found to be significantly impacted (*p* < 0.05). To confirm the results obtained in the present study, further targeted analysis focusing on these pathways is required. It is also known that the number of metabolites of interest, the background set, and the chosen pathway-based library make a substantial difference to the final results [45].

Ketoleucine was present in the pathways identified either by enrichment analysis (branched-chain amino acid catabolism) or pathway analysis (valine, leucine, and isoleucine biosynthesis; *p* = 0.07). Leucine, a proteinogenic branched-chain amino acid (BCAA), is used in humans for energy production, and the branched-chain α-keto acid (BCKA) ketoleucine is one of the first products of leucine breakdown. In the study conducted by Elliott et al. [54], the authors suggested that people who exercise, and therefore have more substantial energy needs, present more enzymatic activity for breaking down leucine into ketoleucine. Another study, which aimed to assess changes in adipose tissue and the activity of enzymes related to BCAA catabolism in early-lactation dairy cows [55], reported that the abundance of adipose tissue has a role in the reversible degradation of BCAA to BCKA. Other authors, such as Eckel et al. [16] and Dervishi et al. [17], found alterations in the concentration of amino acids in the urine and serum metabolomes, respectively, during dairy cows’ transition period. Ruminants obtain BCAA from ruminal microbial activity and supplementation since cows cannot synthesize them. However, they play an important role in many physiological processes that contribute to lameness development when imbalanced, such as skeletal muscle growth stimulation, metabolic homeostasis in adipose tissue maintenance, and the improvement of immune performance [56,57].

In this study, D-gluconolactone was also present in the pathways identified by the enrichment and pathway analysis, but its role in immunity and inflammation is more difficult to establish. According to the Human Metabolome Database [58], gluconolactone can be produced by the enzymatic oxidation of D-glucose. In humans, glucose-6-phosphate dehydrogenase deficiency is a common genetic polymorphism characterized by a defect in the glucose-6-phosphate dehydrogenase gene [59]. Zachut et al. [60] proposed milk glucose-6-phosphate (G6P) as a potential indicator of energy balance in early-lactation dairy cows due to the significant correlation between milk glucose-6-phosphate dehydrogenase (G6PDH) activity and G6P content. More recently, the same research group [61] found a similar correlation and proposed G6PDH as a candidate indicator of negative energy balance in postpartum cows. G6PDH has also been used as a biomarker of oxidative stress in dairy cattle [62], experienced during the transition period [63], and as a target for cancer treatment due to its potential oncogenic activity [64,65]. In the ribose-5-phosphate isomerase deficiency pathway, D-gluconolactone was also present. In humans, ribose-5-phosphate isomerase is encoded by the RPIA gene, and its deficiency is a genetic disorder caused by its mutations; RPIA is, in turn, involved in the pentose phosphate pathway (PPP) as part of carbohydrate degradation. In dairy cows, gene expression levels of RPIA were considerably higher in animals susceptible to endometritis than in resistant ones [66]. It is also known that G6P is catabolized by the PPP, which generates ribose-5-phosphate [67]. In the transaldolase (TAL) deficiency pathway, the D-Gluconolactone metabolite is again present. This pathway is also related to the PPP, which is demonstrated by Perl [68]. The author states that, to understand the pathogenesis of TAL deficiency, it is important to understand the role TAL plays in the PPP. The latter pathway was also identified after performing pathway analysis combining ORA and topology analysis. The connection between the PPP, the TAL-deficient pathway, and the ribose-5-phosphate isomerase deficiency pathway suggests that they have a relevant role in inflammation, and to understand the process they should not be interpreted individually. To our knowledge, there are no studies relating TAL deficiency to transition dairy cows. D-gluconolactone was identified as relevant in the group of samples collected post-calving (POST), which makes it an important metabolite with the potential to be used as a biomarker. Around the time of calving, the transition period, there is an increased risk of subclinical disorders and systemic inflammation after parturition [1,69]. Herzberg et al. [70] also performed LC-MS/MS, but for a proteomic profile of spinal cord samples of chronically lame dairy cattle. The authors found a relevant upregulation of interacting proteins with chaperone and stress functions associated with the glycolytic pathway, and also a downregulation function of myelin basic protein in lame cows. They additionally verified that a considerable number of proteins were involved in the pentose phosphate pathway.

Valproic acid was another metabolite of interest that was an observed hit in the valproic acid metabolism pathway (*p*-value 0.017) and valproic acid pathway (*p*-value 0.034). In humans, valproic acid is established as a drug to treat epilepsy [71]. In the serum of high-yielding dairy cows, this metabolite was reported to be upregulated on the twenty-sixth day post-parturition after zeolite clinoptilolite supplementation [72]. The authors speculate that collective interactions between metabolites, including valproic acid and correlated proteins revealed after joint pathway and interaction analysis, may restore negative energy balance or infection. The relationship between the findings of these authors and the role of valproic acid in immunity and inflammation caused by lameness is not immediately apparent. In the future, the role of valproic acid needs to be further investigated.

Zhang et al. [15] and Dervishi et al. [17], using targeted metabolomics, reported that transition cows affected with lameness after calving had alterations in several innate immunity serum reactants before the appearance of clinical symptoms that could be detected by visual inspection, allowing the monitoring of health status in the near future. These authors recognized the need for further confirmation with larger cohorts for validation. The conclusion from the study conducted by Eckel et al. [16] was that the metabolomics approach has the potential to distinguish lame from healthy cows. These authors also acknowledged that their study should be validated using a larger set of samples. More recently, Randall et al. [1], using untargeted metabolomics, also concluded that further work was needed to confirm metabolite identification followed by the pathway analysis of predictive metabolites, since a better understanding of lameness pathogenesis will allow for the development of strategies for lameness prevention and management in dairy cattle. The results obtained in the current experiment demonstrate that determining a reliable set of metabolites to identify diseased animals, alongside accurate metabolite annotation, is crucial to understanding the roles of metabolites in the biological pathways of dairy cattle lameness. The identification of metabolites is essential for the subsequent pathway analysis.

### 4.5. Usefulness of Metabolite Detection in Urine of Transition Dairy Cows

Though the analysis of metabolite biomarkers in veterinary samples is currently rare, the collection of urine samples in the field is non-invasive and practically feasible. The feasibility of this approach for diagnostics was demonstrated in one study conducting analysis of metabolites called grayanotoxins, which are cardiotoxic and arise from the ingestion of plants, using LC-MS/MS on samples from a veterinary practice. Analysis was considered rapid, quantitative, and specific, and therefore is especially suited to veterinary diagnostic laboratory circumstances when a rapid diagnosis is needed [73]. The analysis of biomarkers of lameness in cows would have similar advantages. Future cost–benefit analysis is needed to evaluate whether the cost of regular sampling and testing for the early detection of lameness outweighs the cost of late-detected lameness to farmers. Understanding the pathology of lameness was another aim of our study, and this understanding could enable better future treatments. Research on the potential use of biomarkers for diagnosis has been increasing, making the topic relevant to be considered in the training of veterinarians [74], and arguably also technicians and farmers [75].

### 4.6. Study Limitations

The reliability of the compiled list of important metabolites was a concern that was mitigated by the use of a combination of different analytic methods [1], such as ML, including five models, a stabiliser, and triangulation to increase the confidence in the selection of the final set of metabolites. However, the study also highlights the limitations of the compound annotation and pathway analysis of untargeted metabolomics.

As Sumner et al. [25] advised in ‘Proposed minimum reporting standards for chemical analysis’, publications should clearly report the level of identification for all listed metabolites. Theodoridis et al. [26] state that findings can only be validated and potentially translated into biological functions after identification based on a minimum of two independent and orthogonal data (analytical attributes) matched to an authentic compound (standard) analysed on the same instrument under the same analytical conditions (i.e., L1). In mass spectrometry, one of these is a mass spectrum of the fragmented metabolite.

Despite advances in detecting compounds due to improved instrument sensitivity, results show that matching mass spectra remains challenging [26,76], and also that peak annotation and identification, especially for untargeted metabolomics, is still a major issue [20,26,77]. Even high-resolution MS is insufficient for isomer identification [22,76], and putative molecular ions need to be checked to ensure that they are not adducts or in-source fragments [26] before putative annotations are searched for based on *m*/*z*. Metabolomes of some specimens are not yet fully described and catalogued; therefore, the metabolites that can be expected in these samples are unknown, and implausible annotations might be made by researchers lacking chemical standards and orthogonal data [26].

The sample protocol relied on a cohort from a single herd already established in a previous study [1], which is not ideal. A larger and more diverse sample would reinforce the results obtained and allow their generalization. Furthermore, for this study, no standards were purchased. In any case, it is not always possible to acquire all the necessary standards, since in most cases they are either not available on the market or prohibitively expensive. With more authentic standards of MS/MS spectra, it would be possible to build a larger library, which would potentially increase success when comparing the m/z and fragmentation spectra of the QC.RAW data files.

## 5. Conclusions

The findings from this study demonstrate that metabolites in lesser-studied samples such as cow urine can be identified with the highest accuracy using chemical standards, RT, and MS/MS data. However, it was difficult to identify all ions due to the lack of matches in *m*/*z* databases, leading to no standards being selected, a lack of matches in MS/MS databases, unavailability and the high cost of buying standards, chimeric MS/MS spectra, and some metabolites not fragmenting. Several limitations common with other studies involving untargeted LC-MS/MS analysis are pointed out and discussed in Section 4.6; in our study specifically, we consider that the limited size of the standards library may have contributed to reducing the number of metabolites annotated with a high level of confidence. Despite these challenges, we were able to identify three significant metabolites at the highest confidence level as predictive of lameness in dairy cows, and thus give biological insight into the pathology of this disease. The enrichment and pathway analysis identified six pathways significantly impacted by lameness, despite the few metabolites identified, giving a better biological understanding of a previously reported lameness prediction model. Further validation using a larger and more diverse sample is needed to confirm the results obtained in this study. Future work to increase the level of confidence in metabolite identification could involve using NMR spectroscopy or a combination of methods such as LC-NMR-MS. Future research may also evaluate the practicality and cost of integrating the proposed approach into routine clinical practice in the national herd. This research would also benefit from including a larger and more diverse sample to assess the generalization of the obtained results.

## Figures and Tables

**Figure 1 animals-14-02030-f001:**
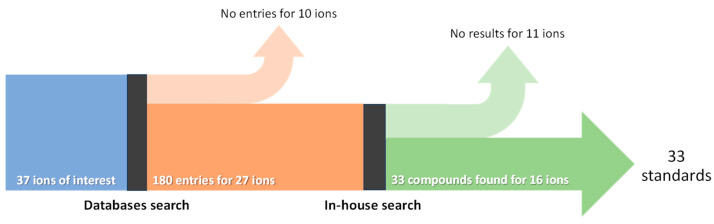
Sankey diagram showing the process for choosing the authentic standards to include in the LC-MS/MS analysis.

**Figure 2 animals-14-02030-f002:**
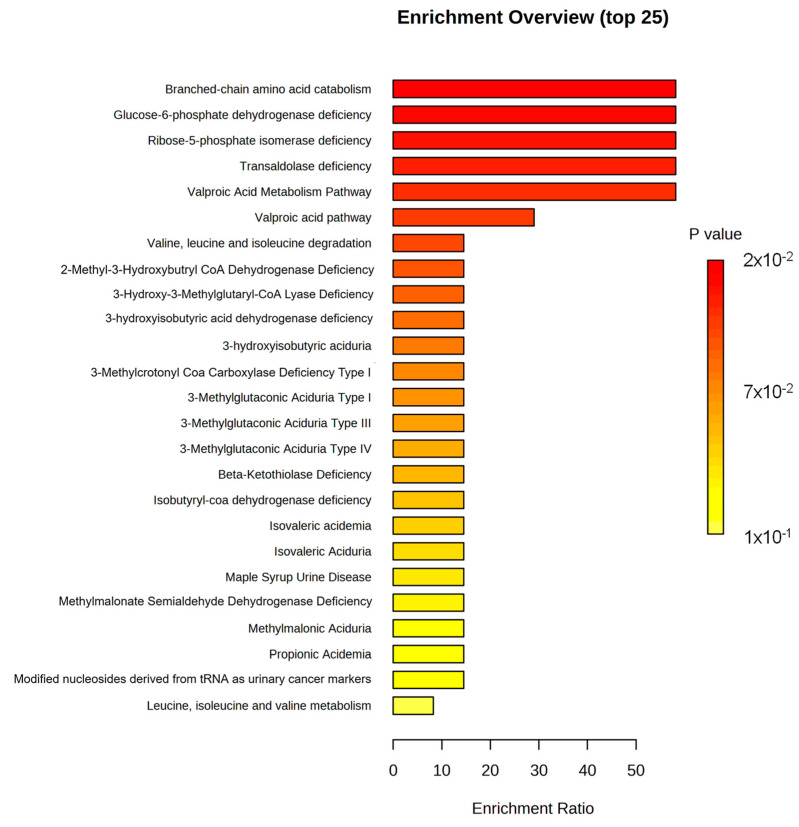
Summary plot for ORA (adapted from Metaboanalyst 5.0).

**Figure 3 animals-14-02030-f003:**
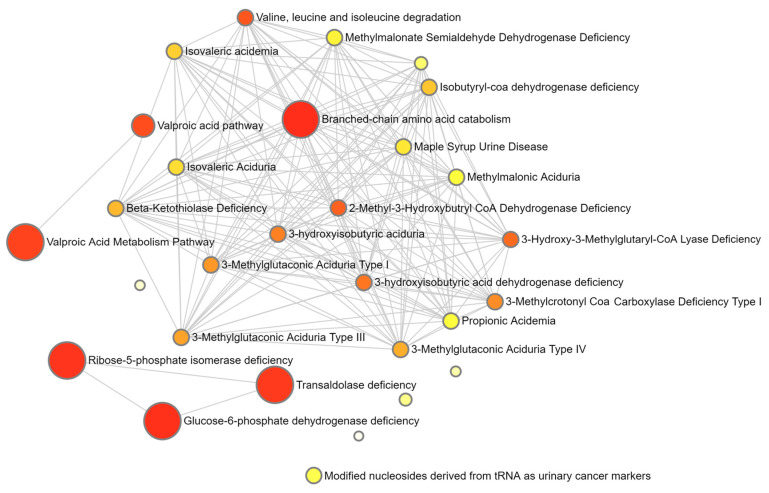
Network view of all enriched pathways (from Metaboanalyst 5.0). Each node (circle) within the network represents a set of metabolites. The colour and size of the nodes are based on their *p*-value and fold enrichment, respectively. Two nodes are connected by a line when they share more than 20% of metabolites.

**Table 1 animals-14-02030-t001:** List of mass ions selected by machine learning models and stability selection from LC-MS data generated using untargeted metabolomics on urine samples of dairy heifers. The urine samples were collected at three different time points (PRE, POST, and AT).

Mass [*m*/*z*]	Name	Formula	Species	Polarity	RT ^1^ [min]	Type of Identification	MSI ^2^ Classification of Confidence in Identification	Sample Collection Time
91.0541	Unknown	Unknown	[M+H]+1	Positive	4.90	NA	L4	AT
107.0500 *	4-Methylphenol	C7 H8 O	[M−H]−1	Negative	4.44	Accurate mass, MS/MS	L2	AT
120.1019	Unknown	Unknown	[M+H]+1	Positive	6.79	NA	L4	POST
129.0561 *	Ketoleucine	C6 H10 O3	[M−H]−1	Negative	8.24	Accurate mass	L3	POST
133.1335	Unknown	Unknown	[M+H]+1	Positive	8.17	NA	L4	AT
143.1080	Valproic acid	C8 H16 O2	[M−H]−1	Negative	3.87	Accurate mass	L3	AT
151.0613 *	Unknown	Unknown	[M+H]+1	Positive	6.75	NA	L4	PRE
158.0811	CEGABA	C7 H13 N O4	[M+H−H_2_O]+1	Positive	6.94	Accurate mass	L3	AT
163.0963	Unknown	Unknown	[M+H]+1	Positive	7.00	NA	L4	AT
174.0562	Indole-2-acetic acid	C10 H9 N O2	[M−H]−1	Negative	7.16	Accurate mass	L3	AT
177.0406	D-Glucono-delta-lactone	C6 H10 O6	[M−H]−1	Negative	7.57	Accurate mass	L3	POST
193.0700	Unknown	Unknown	[M−H]−1	Negative	6.29	NA	L4	PRE
195.0664 *	AH0675000	C10 H12 O4	[M−H]−1	Negative	7.55	Accurate mass	L3	PRE
195.1225	Unknown	Unknown	[M+H]+1	Positive	4.76	NA	L4	PRE
199.1440	Unknown	Unknown	[M+H]+1	Positive	6.36	NA	L4	AT
201.0771	Unknown	Unknown	[2M−H]−1	Negative	7.66	NA	L4	POST
201.1596 *	Unknown	Unknown	[M+H]+1	Positive	6.62	NA	L4	PRE
203.0021	Unknown	Unknown	[M−H]−1	Negative	7.07	NA	L4	AT
215.0203 *	Unknown	Unknown	[M−H]−1	Negative	7.39	NA	L4	POST
216.0514	Unknown	Unknown	[M−H]−1	Negative	8.31	NA	L4	POST
229.9764	Unknown	Unknown	[M−H]−1	Negative	8.28	NA	L4	AT
243.1236	Unknown	Unknown	[2M−H]−1	Negative	6.83	NA	L4	AT
251.0771 *	Unknown	Unknown	[M+H]+1	Positive	7.05	NA	L4	POST
256.1751	Unknown	Unknown	[M+NH_4_]+1	Positive	4.45	NA	L4	PRE
280.1385	Unknown	Unknown	[M+H]+1	Positive	7.12	NA	L4	PRE
283.1034	1-Methylinosine	C11 H14 N4 O5	[M+H]+1	Positive	6.75	Accurate mass	L3	PRE
290.1229	Unknown	Unknown	[M+H]+1	Positive	7.91	NA	L4	POST
291.0180 *	Unknown	Unknown	[M−H]−1	Negative	6.65	NA	L4	POST
299.0229	Unknown	Unknown	[M−H]−1	Negative	3.58	NA	L4	POST
301.0748	Unknown	Unknown	[M−H]−1	Negative	3.36	NA	L4	PRE
312.1297	N2-Dimethylguanosine	C12 H17 N5 O5	[M+H]+1	Positive	6.69	Accurate mass	L3	PRE
314.1593	Unknown	Unknown	[M+H]+1	Positive	4.23	NA	L4	AT
316.1384	Unknown	Unknown	[M+H]+1	Positive	5.03	NA	L4	POST
322.0930	Unknown	Unknown	[M−H]−1	Negative	4.85	NA	L4	AT
489.1922	Unknown	Unknown	[M+H]+1	Positive	8.70	NA	L4	PRE
661.2039	Unknown	Unknown	[M−H]−1	Negative	6.41	NA	L4	PRE
881.2806	Unknown	Unknown	[M+H]+1	Positive	7.82	NA	L4	AT

* Mass ions selected by both stabiliser and machine learning models. ^1^ RT—retention time; ^2^ MSI—Metabolomics Standards Initiative.

**Table 2 animals-14-02030-t002:** List of authentic standards used for metabolite identification.

Target *m*/*z* (i.e., the *m*/*z* of the Metabolite in the Inclusion List)	Standards		
	Adduct	Polarity	Name
107.0500	M−H	Negative	*p*-Cresol (4-methylphenol)
120.1019	M+NH_4_	Positive	2-Methybutyric acid
120.1019	M+NH_4_	Positive	Ethyl propionate
120.1019	M+H	Positive	*N*-Methyldiethanolamine
120.1019	M+NH_4_	Positive	Pivalic acid
120.1019	M+NH_4_	Positive	Valeric acid
129.0561	M−H	Negative	Acetylbutyric acid
133.1335	2M+H−H_2_O	Positive	1-Amino-2-propanol
133.1335	M+NH_4_−H_2_O	Positive	Diisopropylamine
143.1080	M−H	Negative	Valproic acid
151.0613	M+IsoProp+H	Positive	Oxalate
174.0562	M−H	Negative	3-Indoleacetic acid
177.0406	M−H_2_O−H	Negative	D-Gluconic acid
177.0406	M−H	Negative	D-Glucono-1,4-lactone
177.0406	M−H	Negative	Gluconolactone
177.0406	M+FA−H	Negative	Glutaric acid
177.0406	M+FA−H	Negative	Methyl succinate
177.0406	M+Hac−H	Negative	Methylmalonic acid
177.0406	M+Hac−H	Negative	Succinic acid
195.0664	M−H	Negative	Dimethoxyphenylacetic acid
195.0664	M+FA−H	Negative	Hydroxycoumarin
195.0664	M+Hac−H	Negative	Phenylacetic acid
195.0664	M+Hac−H	Negative	Toluic acid
195.0664	M+FA−H	Negative	Tolylacetic acid
195.1225	M+IsoProp+H	Positive	Deoxyribose
215.0203	M+FA−H	Negative	Gallate
251.0771	M+H−H_2_O	Positive	Inosine
283.1034	2M+H−H_2_O	Positive	D-Arabinose
283.1034	2M+H−H_2_O	Positive	D-Ribose
290.1229	M+NH_4_	Positive	Arbutin
312.1297	M+IsoProp+Na+H	Positive	Deoxyuridine
316.1384	2M+2H+3H_2_O	Positive	1-Naphthol
316.1384	2M+2H+3H_2_O	Positive	2-Naphthol

**Table 3 animals-14-02030-t003:** Putative annotations and Metabolomics Standards Initiative (MSI) classification of confidence in identification. Valproic acid standard did not fragment. Some ions had multiple chromatographic peaks identified with the same name, which were all important metabolites.

Mass [*m*/*z*]	Name	Type of Identification	MSI Classification of Confidence in Identification
174.05623	Arbutin	NA	NA
291.01797	Arbutin	NA	NA
107.05003	Cresol	Accurate mass, RT, MS/MS	L1
243.12357	Diethylene Glycol	NA	NA
216.05141	Gluconolactone	NA	NA
177.04063	Gluconolactone	Accurate mass, RT, MS/MS	L1
195.06644	Gluconolactone	NA	NA
243.12357	Glycerol	NA	NA
143.10802	Methylpentanoic acid	NA	NA
143.10802	Valproic acid	Accurate mass, RT	L2

## Data Availability

The original data presented in the study will be openly available after acceptance in the Nottingham Research Data Management Repository at https://rdmc.nottingham.ac.uk/.

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
