# Peer review of "Identification of Predictive Biomarkers of Lameness in Transition Dairy Cows"

_animals, 2024, doi:10.3390/ani14142030_

Round 1

Reviewer 1 Report

Comments and Suggestions for Authors

ID: animals-3049091

Identification of predictive biomarkers of lameness in transition dairy cows

The authors aimed to investigate the metabolites previously determined as predictors of lameness in the transition period and to understand their biological role by conducting pathway analysis. Metabolic pathway analysis allows us to understand the interactions between metabolism and immunity in lameness. They used predictive biomarkers and machine learning.

The “Simple Summary” and “Abstract” are well written.

The “Introduction” section is well but need some improvements as mentioned below.

The “Materials and Methods” section is well written and enough to understand. But the ML should be given detailed.

The “Results” section is adequate for the aim but need major revisions mentioned below.

The “Discussion” sections should be re-written there is no real academic discussion, this section is too weak, such as there is no info about the accuracy.

The “Study limitations” are real.

The “Conclusions” section is nor supported by the results and discussion.

Line 17: “highest confidence level”. No proof in the text.

Line 30: “p-value” -> “p”. No need to write “value”.

Lines 39-59: Please mention the major causes of lameness at the first paragraph.

Line 71: which ML algorithms? ML is a label of a class.

Please mention the accuracy of literature [11,24] in the text.

Line 117: “67 Holstein dairy heifers”. Please give a proof that this sample size is adequate for this aim.

Line 126: “using R” Please mention the library name(s).

Line 202: Please give all statistical details about ML.

Line 208: “28 (twenty-eight)” à “28” is enough.

Line 246: The Figure 2 should be transformed and re-organized as Table to be clearer.

Line 248: Figure 3: Please define the means of colors.

This manuscript must be rejected because of many lacks.

Author Response

General comments

The authors aimed to investigate the metabolites previously determined as predictors of lameness in the transition period and to understand their biological role by conducting pathway analysis. Metabolic pathway analysis allows us to understand the interactions between metabolism and immunity in lameness. They used predictive biomarkers and machine learning.

We thank the reviewer for their assessment of our work. Please see the ensuing replies for details on how we addressed each specific comment. All changes made to the text are highlighted in blue on the manuscript.

The “Simple Summary” and “Abstract” are well written.

We thank the reviewer for the positive comment.

The “Introduction” section is well but need some improvements as mentioned below.

We thank the reviewer for their suggestions - the introduction has been improved in response to the reviewer comments.

The “Materials and Methods” section is well written and enough to understand. But the ML should be given detailed.

The paper has been updated to include the suggested details. To clarify, all the information about the ML analysis had already been published in (Randall et al., 2023), as cited in the manuscript. We only used the list of mass ions selected by ML models and stability selection, from LC-MS data previously generated. Please see the responses below for more details.

The “Results” section is adequate for the aim but need major revisions mentioned below.

The Results section has been updated to include the suggested details mentioned.

The “Discussion” sections should be re-written there is no real academic discussion, this section is too weak, such as there is no info about the accuracy.

In response to the reviewer’s recommendation, we have made a major revision of the Discussion section 4.4. (Enrichment & pathway analysis) and a new section (4.5. Usefulness of metabolite detection in urine of transition dairy cows) was added. We are not entirely sure what the reviewer is referring to when they mention accuracy, but assuming this comment concerns the ML analysis, these details, as mentioned previously and explained in the manuscript, were published previously in (Randall et al. 2023).

The “Study limitations” are real.

We thank the reviewer for their comment. We have added a new paragraph at the end of this section about the size of the standard library we used (Line 476 – 481): “With more authentic standards MS/MS spectra it would be possible to build a larger library, which would potentially increase the success when comparing m/z and fragmentation spectra of the QC.RAW data files.”

The “Conclusions” section is nor supported by the results and discussion.

We thank the reviewer for their remark; however, in our opinion, the conclusion is consistent with the results we obtained whilst also highlighted the limitations of the study. Additional sentences about the limitations (Line 488 – 491) and future research were added at the end (Line 500 – 503).

Line 17: “highest confidence level”. No proof in the text.

Thank you for this comment, if by proof the reviewer is referring to the definitions of highest confidence and how this relates to our results, then section 2.2. (Previous metabolite annotation and identification) offers this information (Line 171 – 175).  It is described here that levels 1 to 4 are based on the recommendation of the Chemical Analysis Working Group, Metabolomics Standards Initiative (MSI). When we refer to the highest confidence level, we mean Level 1 as also explained in the first paragraph of the Results section (“…classified as L1, the highest level of identification…”). We have also included a summary of these results in the abstract, for improved clarity. We hope this addresses the reviewer’s concern.

Line 30: “p-value” -> “p”. No need to write “value”.

Thank you, we have changed “p-value” to “p” as suggested by the reviewer (Lines 36 and 37).

Lines 39-59: Please mention the major causes of lameness at the first paragraph.

We thank the reviewer for this suggestion – a brief description of these causes has been added (Line 47– 49).

Line 71: which ML algorithms? ML is a label of a class.

We thank the reviewer for this comment and – a description of all the algorithms has been included (Line 85 – 88).

Please mention the accuracy of literature [11,24] in the text.

As requested, we have added the accuracies reported in the literature.

Line 117: “67 Holstein dairy heifers”. Please give a proof that this sample size is adequate for this aim.

In response to the reviewer’s request, we have clarified the sample size and added the justification for using that sample size – “Cohort size was based on a pilot work published by He et al. [29] in dairy cows using a similar statistical analysis.”. However, we would like to highlight that, as mentioned in the manuscript, we are using LC-MS results generated in a previously published study; this study was not generating new data.

Line 126: “using R” Please mention the library name(s).

We thank the reviewer for pointing that out, and we have added further detail and two references (Line 149):

  1. Kuhn [aut, M.; cre; Wing, J.; Weston, S.; Williams, A.; Keefer, C.; Engelhardt, A.; Cooper, T.; Mayer, Z.; Kenkel, B.; et al. Caret: Classification and Regression Training 2022.
  2. R: The R Project for Statistical Computing Available online: https://www.r-project.org/ (accessed on 27 May 2024).

Line 202: Please give all statistical details about ML.

As mentioned above, the data was generated and ML analysis was conducted in a previous study (Randall et al 2023), where all statistical details are reported. As this was not part of the current study the details have not been reported, however, we have referenced these in Lines 84, 148, 154, 285 and 453. For the current study we used the list of mass ions selected by ML models and stability selection. We have added additional information in sections ”1. Introduction” and “2.1. Previous generation of LC-MS data” to also make this clearer to the reader.

Line 208: “28 (twenty-eight)” à “28” is enough.

We have deleted “twenty-eight”, as requested by the reviewer.

Line 246: The Figure 2 should be transformed and re-organized as Table to be clearer.

We thank the reviewer for their feedback. However, we believe that the plot included is the most appropriate way to present the results generated, and this plot is a standard graph generated by the Metaboanalyst software, commonly used by researchers in the field of metabolomics. This type of graph has been consistently used in publications in many journals from MDPI and other publishers, as can be seen in the following examples:

https://www.mdpi.com/2218-1989/9/10/207

https://link.springer.com/article/10.1007/s11306-021-01770-x

https://www.sciencedirect.com/science/article/pii/S0753332220307678?via%3Dihub

To make this plot clearer we modified it slightly so that the reader may have a better and more immediate view of the p<0.05 threshold and which pathways are within that threshold.

Line 248: Figure 3: Please define the means of colors.

We thank the reviewer for pointing this omission out – a complete description of the meaning of the colours and circle sizes was added to the figure caption.

This manuscript must be rejected because of many lacks.

We thank the reviewer for all the recommendations and for pointing out weaknesses that should be improved. We have made major changes throughout the manuscript to address the comments made by all the reviewers. We believe that the manuscript is appropriate for publication and welcome any further comments from reviewers.

Reviewer 2 Report

Comments and Suggestions for Authors

The paper investigates the identification of predictive biomarkers for lameness in transition dairy cows using liquid chromatography-tandem mass spectrometry (LC-MS/MS). The main contributions include the accurate identification of three metabolites (cresol, valproic acid, and gluconolactone) as predictors of lameness, and the analysis of their involvement in significant biological pathways. This study enhances the understanding of the interactions between metabolism and immunity in lameness and proposes early detection methods to improve dairy cow welfare and farm economics.

The paper fits well within the scope of the journal as it addresses significant challenges in animal welfare and agricultural sustainability by exploring advanced diagnostic techniques in veterinary science.

The paper effectively addresses an important issue in dairy farming by exploring metabolic biomarkers for early lameness detection. However, several areas require improvement:

The main research question is whether specific metabolites in urine can predict lameness in dairy cows during the transition period.

The topic is both original and relevant, addressing a critical gap in early lameness detection in dairy cows, which has significant implications for animal welfare and farm economics.

This research adds to the existing body of knowledge by providing a high-confidence identification of lameness biomarkers and linking these to specific metabolic pathways, offering potential for improved early detection methods.

Simple Summary Revision:

The simple summary should be rewritten to better contextualize the paper within existing literature, using non-technical language to ensure accessibility to a broader audience.

Abstract:

The abstract correlates with the manuscript content but could be improved by including more specific results and their significance. The abstract should be rewritten to include more detailed results and highlight the significance of the findings.

Introduction:

Lines 43: regarding early culling cows I suggest read and cite: 10.3389/fvets.2023.1141286

I suggest the authors provide examples from other studies that have identified early biomarkers of disease using urine and blood samples. Specifically, including examples related to general health and ruminal acidosis would help contextualize your findings within the broader literature on early disease detection in dairy cows. This will highlight the relevance and innovation of your approach in identifying metabolic biomarkers for lameness. See 10.29261/pakvetj/2020.067 and 10.1111/jpn.13607.

MethodologY:

The methodology could be strengthened by increasing the sample size and including a broader range of environmental variables to enhance the robustness of the findings. Additionally, employing a more comprehensive validation process with different cohorts could improve the generalizability of the results.

Discussion:

Future research should focus on validating these findings with larger and more diverse cohorts. Exploring additional biomarkers and their integration into predictive models could further enhance early detection systems. Additionally, investigating the economic impact of implementing such detection methods on a larger scale would be beneficial.

Consider discussing the potential economic implications of early lameness detection in more detail.

Explore the integration of identified biomarkers into practical diagnostic tools for farmers.

I recommend incorporating a discussion paragraph highlighting the significance of these new findings regarding predictive biomarkers of lameness in transition dairy cows in university courses for educating future veterinarians, technicians, and farmers about these issues. Effective teaching methods are crucial for shaping knowledgeable students and proficient veterinarians. Refer to recent publications on veterinary education to provide up-to-date insights into best practices for preparing future professionals to address the challenges discussed in the paper. Specifically, consider incorporating insights from studies such as 10.1016/j.jevs.2023.104537 and 10.3390/ani13223503 to support your arguments.

Conclusions:

The conclusions are generally consistent with the presented evidence. However, they would benefit from a more detailed discussion on the limitations and potential confounding factors.

References:

The references are appropriate and relevant, but ensuring all cited works are directly mentioned in the text would enhance credibility.

Ensure all references cited in the manuscript are included in the reference list and vice versa.

Author Response

Comments 1: General comments

The paper investigates the identification of predictive biomarkers for lameness in transition dairy cows using liquid chromatography-tandem mass spectrometry (LC-MS/MS). The main contributions include the accurate identification of three metabolites (cresol, valproic acid, and gluconolactone) as predictors of lameness, and the analysis of their involvement in significant biological pathways. This study enhances the understanding of the interactions between metabolism and immunity in lameness and proposes early detection methods to improve dairy cow welfare and farm economics.

The paper fits well within the scope of the journal as it addresses significant challenges in animal welfare and agricultural sustainability by exploring advanced diagnostic techniques in veterinary science.

The paper effectively addresses an important issue in dairy farming by exploring metabolic biomarkers for early lameness detection. However, several areas require improvement:

The main research question is whether specific metabolites in urine can predict lameness in dairy cows during the transition period.

The topic is both original and relevant, addressing a critical gap in early lameness detection in dairy cows, which has significant implications for animal welfare and farm economics.

This research adds to the existing body of knowledge by providing a high-confidence identification of lameness biomarkers and linking these to specific metabolic pathways, offering potential for improved early detection methods.

Response 1: We thank the reviewer for their positive comments regarding the importance of the work conducted. Please see the ensuing responses for details on how we addressed each specific comment. All changes made to the text are highlighted in blue on the manuscript.

Comments 2: Simple Summary Revision:

The simple summary should be rewritten to better contextualize the paper within existing literature, using non-technical language to ensure accessibility to a broader audience.

Response 2: We thank the reviewer for their feedback. we have addressed the concerns by revising the simple summary.

Comments 3: Abstract

The abstract correlates with the manuscript content but could be improved by including more specific results and their significance. The abstract should be rewritten to include more detailed results and highlight the significance of the findings.

Response 3: We thank the reviewer for this insightful suggestion – the levels of confidence in identification, which are the main results of this study, were now included (Line 34). Furthermore, we incorporated the sentence: “This approach shows potential as a reliable method to identify biomarkers that can be used in the future to predict the risk of lameness before calving.” at the end of the abstract.

Comments 4: Introduction

Lines 43: regarding early culling cows I suggest read and cite: 10.3389/fvets.2023.1141286

I suggest the authors provide examples from other studies that have identified early biomarkers of disease using urine and blood samples. Specifically, including examples related to general health and ruminal acidosis would help contextualize your findings within the broader literature on early disease detection in dairy cows. This will highlight the relevance and innovation of your approach in identifying metabolic biomarkers for lameness. See 10.29261/pakvetj/2020.067 and 10.1111/jpn.13607

Response 4: The Introduction has been updated to include the suggested details: (A) Line 43 (now line 52): The proposed reference has now been added and cited, (B) although we already had a few examples from other studies that had identified early biomarkers, we have added more references of research that used urine, milk and blood to compare the metabolome of lame cows with control cows (Line 83 – 84). We have also included the two additional references proposed by the reviewer (Lines 81 and 105 respectively).

Comments 5: Methodology

The methodology could be strengthened by increasing the sample size and including a broader range of environmental variables to enhance the robustness of the findings. Additionally, employing a more comprehensive validation process with different cohorts could improve the generalizability of the results.

Response 5: Thank you for your comment - we would like to highlight that, as mentioned in the manuscript, we are using a data set from a previously published study (Randall et al., 2023) and no new data was being generated in the current study, which was solely to annotate the metabolites identified as predictors of lameness in the transition period and understand their role in the pathways and processes that lead to lameness. In response to the reviewer’s suggestion though we have clarified the sample size from the previous study and added the justification for using it – (Line 143 – 144): “Cohort size was based on a pilot work published by He et al. [29] in dairy cows using a similar statistical analysis”.

We agree that future work and more diverse sample to confirm the results obtained in this study and provide external validation. A statement to reflect this has been added to the “4.5. Study limitations” (Line 475 – 476) and the Conclusion (Line 497).

Comments 6: Discussion

Future research should focus on validating these findings with larger and more diverse cohorts. Exploring additional biomarkers and their integration into predictive models could further enhance early detection systems. Additionally, investigating the economic impact of implementing such detection methods on a larger scale would be beneficial.

Consider discussing the potential economic implications of early lameness detection in more detail.

Explore the integration of identified biomarkers into practical diagnostic tools for farmers.

I recommend incorporating a discussion paragraph highlighting the significance of these new findings regarding predictive biomarkers of lameness in transition dairy cows in university courses for educating future veterinarians, technicians, and farmers about these issues. Effective teaching methods are crucial for shaping knowledgeable students and proficient veterinarians. Refer to recent publications on veterinary education to provide up-to-date insights into best practices for preparing future professionals to address the challenges discussed in the paper. Specifically, consider incorporating insights from studies such as 10.1016/j.jevs.2023.104537 and 10.3390/ani13223503 to support your arguments.

Response 6: Thank you for these comments. In response to the reviewer’s recommendation, we have added a new paragraph between “4. Discussion” and “4.1. Metabolite annotation and identification” sections. A major revision of the Discussion section 4.4. (“Enrichment & pathway analysis”) and a new section (4.5. Usefulness of metabolite detection in urine of transition dairy cows) was also added to the Discussion where the two proposed references were cited.

Comments 7: Conclusions

The conclusions are generally consistent with the presented evidence. However, they would benefit from a more detailed discussion on the limitations and potential confounding factors.

Response 7: We thank the reviewer for their comment – although the conclusion already pointed out a few limitations, we improved the writing and highlighted the challenge regarding the standard library size (Line 489 – 491), which was also added to the Discussion section “Study limitations” (Line 476 – 481). A few additional sentences clarifying future research were also added at the end of the Conclusions section (Line 500 – 503).

Comments 8: References

The references are appropriate and relevant, but ensuring all cited works are directly mentioned in the text would enhance credibility.

Ensure all references cited in the manuscript are included in the reference list and vice versa.

Response 8: We thank the reviewer for pointing out the importance of the consistency between the cited works and reference list. References have been checked to ensure no mismatching.

Reviewer 3 Report

Comments and Suggestions for Authors

General information:

·        There is no doubt that the authors have made a great effort to produce good scientific work.

·        Interesting aspects were described in lines 10-12: Early detection of lameness allows new cases to be treated and relapses to be avoided. This is crucial to minimize economic losses and animal pain, thus improving cow welfare and the sustainability of the dairy industry. Unfortunately, this goal cannot be achieved within the scope of this work.

·        In line 34, the authors stated that they were unsure whether to disseminate the results and therefore wrote: To assess the generalizability of these results, validation with a larger cohort is needed.

Some comments on the manuscript:

·       In line 117, the authors did not mention whether they took one urine sample from each cow or two samples, one before and one after calving. To give us a good overview, there is no urine sampling plan included in the work.

·       The urine samples were described in lines 116-121. Unfortunately, it was not clear whether there were cows that were ill during the period when the urine samples were taken or whether some of them became ill later.

·       Under “Materials and Methods” there are topics that are incorrectly numbered. This needs to be corrected.

·       In line 271 it was mentioned that significantly different p-cresol values ​​were also observed in the cerumen samples between lame and control calves. The question is: Did the calves become ill because of their mother’s illness or did they become ill later?

·       The reference list is fine. However, there are a few minor things that need to be corrected.

  1. Reference 2 was not written in the journal line.
  2. Reference numbers 25, 26, 27 and 37, publication dates are not in bold.

Author Response

Comments 1: General comments

- There is no doubt that the authors have made a great effort to produce good scientific work.

- Interesting aspects were described in lines 10-12: Early detection of lameness allows new cases to be treated and relapses to be avoided. This is crucial to minimize economic losses and animal pain, thus improving cow welfare and the sustainability of the dairy industry. Unfortunately, this goal cannot be achieved within the scope of this work.

- In line 34, the authors stated that they were unsure whether to disseminate the results and therefore wrote: To assess the generalizability of these results, validation with a larger cohort is needed.

Response 1: We thank the reviewer for their assessment of our work. This work was the second step towards early detection of lameness after identifying that there was a metabolic difference between lame and non-lame cows. We identified some of the metabolites with a high level of confidence. The next steps will be to validate this in another cohort of cows, quantify this, then monitor cows over a period of months/years, then develop a way of applying this test in the field. We do not aim to undertake all steps in this paper. We were sure the identified metabolites should be published and did not mean to give the impression we were unsure about publishing them. The metabolites give us insights into the pathogenesis of lameness through knowing the metabolic pathways affected and these are the findings of this work, which is one step in the research pipeline.

Please see the ensuing replies for details on how we addressed each specific comment. All changes made to the text are highlighted in blue on the manuscript itself.

Comments 2: In line 117, the authors did not mention whether they took one urine sample from each cow or two samples, one before and one after calving. To give us a good overview, there is no urine sampling plan included in the work.

Response 2: In response to the reviewer’s request, we have clarified the sampling protocol under section “2.1. Previous generation of LC-MS data” (Line 141-144). However, we would like to highlight that, as mentioned in the manuscript, we are using LC-MS results generated in a previously published study where sample collection and the cohort are described in detail.

Comments 3: The urine samples were described in lines 116-121. Unfortunately, it was not clear whether there were cows that were ill during the period when the urine samples were taken or whether some of them became ill later.

Response 3: In response to the reviewer’s recommendation, we added the exclusion criteria considered in Randal et al., 2023 research: “Exclusion criteria were surgery or any treatment with antimicrobials or anti-inflammatories that the heifers had been subjected to during the current lactation.” (Line 144 – 146).

Comments 4: Under “Materials and Methods” there are topics that are incorrectly numbered. This needs to be corrected.

Response 4: We thank the reviewer for pointing out this error and this has been addressed.

Comments 5: In line 271 it was mentioned that significantly different p-cresol values ​​were also observed in the cerumen samples between lame and control calves. The question is: Did the calves become ill because of their mother’s illness or did they become ill later?

Response 5: We agree with the reviewer that this is a good question that, in our opinion, the information in the article does not allow us to answer. To address your remark, we added the comment to the manuscript: “However, the authors make a reservation regarding the use of this potential biomarker and point out that more elucidation is needed as it has already been found altered in cattle with other pathological conditions.” (Line 306 – 309).

Comments 6: The reference list is fine. However, there are a few minor things that need to be corrected.

  1. Reference 2 was not written in the journal line.
  2. Reference numbers 25, 26, 27 and 37, publication dates are not in bold.

Response 6: We thank the reviewer for pointing out this mistake – we have sorted the issue concerning reference 2 (now [12]). As for the formatting for dates, we are using Zotero bibliography package implementing the Animals style for referencing, in which books, book sections, software and online material do not show dates highlighted in bold – recommendations and guidelines including these formatting rules can be found in https://www.mdpi.com/journal/animals/instructions#references.

Reviewer 4 Report

Comments and Suggestions for Authors

This study focused on identifying predictive biomarkers of lameness in transitional dairy cows using a metabolomic approach. The study analyzed urine samples from udder-healthy and diseased dairy cows during the time of transition using LC-MS/MS. The authors succeeded in identifying three predictive metabolite markers with high confidence, which is an important finding. Metabolite enrichment analysis identified several important biological pathways associated with the predicted metabolites, providing insights into the interaction between metabolism and immunity in dairy cattle disease. The completeness of the article needs to be supplemented, and some details need to be slightly modified before acceptance.

1. Check the numbering of “2. Materials and methods”, e.g. “4.1. Previous generation of LC-MS data” in line 112 should be “2.1. Previous generation of LC-MS data”.

2. line160-165: Although the authors searched two databases, 11 ions could not be matched with corresponding standards. The biological functions of these unknown metabolites could not be determined, limiting the in-depth understanding of the disease mechanism in dairy cattle. Could the standard library be further expanded to improve the identification rate of unknown metabolites?

3. Add methods to avoid false positive results

4. The authors' analysis of the biological significance of the three predictive metabolites is relatively simple and general. A more in-depth exploration of the specific biological pathways in which these metabolites are involved and their role in the disease process in dairy cattle is absent. It is recommended to add a more in-depth analysis of the biological functions of these metabolites and their relationship with immune regulation.

5. Pathway analyses listed only the pathways associated with metabolites. What are the changes in the expression of key genes and proteins in these pathways, and what is the relationship to the mechanism of claudication.

6. The discussion focused mainly on metabolite identification and biological pathway analysis, and the potential of these findings for clinical application was not discussed. The value of these findings for early detection and prevention of the disease in dairy cattle should be further evaluated.

Author Response

This study focused on identifying predictive biomarkers of lameness in transitional dairy cows using a metabolomic approach. The study analyzed urine samples from udder-healthy and diseased dairy cows during the time of transition using LC-MS/MS. The authors succeeded in identifying three predictive metabolite markers with high confidence, which is an important finding. Metabolite enrichment analysis identified several important biological pathways associated with the predicted metabolites, providing insights into the interaction between metabolism and immunity in dairy cattle disease. The completeness of the article needs to be supplemented, and some details need to be slightly modified before acceptance.

We thank the reviewer for their comments. Please see the ensuing replies for details on how we addressed each specific comment. All changes made to the text are highlighted in blue on the manuscript itself.

Comments 1: Check the numbering of “2. Materials and methods”, e.g. “4.1. Previous generation of LC-MS data” in line 112 should be “2.1. Previous generation of LC-MS data”.

Response 1: We thank the reviewer for pointing out this error which has been addressed.

Comments 2: line160-165: Although the authors searched two databases, 11 ions could not be matched with corresponding standards. The biological functions of these unknown metabolites could not be determined, limiting the in-depth understanding of the disease mechanism in dairy cattle. Could the standard library be further expanded to improve the identification rate of unknown metabolites?

Response 2: We thank the reviewer for this insightful comment – a last paragraph was added at the end of the “4.6. Study limitations” section where we recognise that the limited size of the standards library may have contributed to reducing the success of our experiment (Line 476 – 481). We additionally explained the underlying reasons for the size of the library. This was also mentioned in the conclusion (Line 489 – 491).

Comments 3: Add methods to avoid false positive results

Response 3: We thank the reviewer for this important remark – in section “2.1. Previous generation of LC-MS”, we described the statistical analysis in more detail to address your comment (Line 150 – 154).  We mentioned that stability selection was applied to prevent overfitting and model triangulation was also used to reduce the probability of selecting false positive variables, as Randall et al. described, 2023.

Comments 4: The authors' analysis of the biological significance of the three predictive metabolites is relatively simple and general. A more in-depth exploration of the specific biological pathways in which these metabolites are involved and their role in the disease process in dairy cattle is absent. It is recommended to add a more in-depth analysis of the biological functions of these metabolites and their relationship with immune regulation.

Response 4: In response to the reviewer’s recommendation, we have performed a major revision of the Discussion section 4.4. (“Enrichment & pathway analysis”).

Comments 5: Pathway analyses listed only the pathways associated with metabolites. What are the changes in the expression of key genes and proteins in these pathways, and what is the relationship to the mechanism of claudication.

Response 5: We thank the reviewer for their remark – we addressed their comment by adding more information about branched-chain amino acid (BCAA) expression (Line 365 – 369) and glucose-6-phosphate dehydrogenase (G6PDH) (Line 380 – 382). Additional information was also added concerning the role of ribose-5-phosphate isomerase and RPIA gene (Line 383 – 388). However, from our knowledge, there are no studies relating transaldolase deficiency to transition dairy cows, which has also been explained in the discussion.

Comments 6: The discussion focused mainly on metabolite identification and biological pathway analysis, and the potential of these findings for clinical application was not discussed. The value of these findings for early detection and prevention of the disease in dairy cattle should be further evaluated.

Response 6: We thank the reviewer for their comment – to address this, we have added a new paragraph between the “4. Discussion” and “4.1. Metabolite annotation and identification” sections, and also a new section “4.5. Usefulness of metabolite detection in urine of transition dairy cows”. The sentence “Future research may also evaluate the practicality and cost of integrating the proposed approach into routine clinical practice in the national herd.” was also added in the Conclusion.

Round 2

Reviewer 1 Report

Comments and Suggestions for Authors

The insufficiencies remain after the revision.

Reviewer 2 Report

Comments and Suggestions for Authors

Dear Authors,

I wanted to extend my heartfelt congratulations to you and your team for the outstanding job you've done in revising your paper. I am genuinely impressed by the way you have meticulously incorporated the suggested revisions. Your commitment to improving the article's quality is evident, and I must say that the final result is nothing short of exceptional. The transformation from the initial draft to the current version is remarkable and a testament to your dedication to excellence.

Reviewer 4 Report

Comments and Suggestions for Authors

Thank you very much!